# Ultrasound renal denervation in hypertensive patients: A systematic review and meta-analysis

Roy Novri Ramadhan[1], Derren David Christian Homenta Rampengan[2], Felicia Angelica Gunawan[2], Nathania[1], Sebastian Emmanuel Willyanto[3], Hiroyuki Yamada[4,5], Mochammad Thaha[6,7], Satriyo Dwi Suryantoro[6,7], Maulana Antiyan Empitu[8,9]*

1 Medical Program, Faculty of Medicine, Universitas Airlangga, Surabaya, Indonesia, 2 Medical Program, Faculty of Medicine, Universitas Sam Ratulangi, Manado, Indonesia, 3 Medical Program, Faculty of Medicine, Universitas Brawijaya, Malang, Indonesia, 4 Department of Nephrology, Kyoto University Hospital, Kyoto, Japan, 5 Department of Primary Care and Emergency Medicine, Graduate School of Medicine, Kyoto University, Kyoto, Japan, 6 Department of Internal Medicine, Faculty of Medicine, Universitas Airlangga, Surabaya, Indonesia, 7 Department of Internal Medicine, Universitas Airlangga Hospital, Surabaya, Indonesia, 8 Faculty of Medicine, Airlangga University, Surabaya, Indonesia, 9 Faculty of Health, Medicine and Natural Sciences (FIKKIA), Airlangga University, Banyuwangi, Indonesia

* maulana.antiyan@fk.unair.ac.id

**Data Availability Statement:** "All relevant data are within the paper and its supporting information files."

## Abstract

### Introduction

Hypertension is the leading noncommunicable disease case affecting 1.28 billion individuals worldwide, with most cases located in low- and middle-income countries. While there are numerous techniques for treating mild to moderate hypertension, properly controlling severe or resistant hypertension poses substantial challenges. Ultrasound-based renal denervation (uRDN) has emerged as a promising non-pharmacological approach. This study aims to investigate the safety and efficacy of uRDN in hypertensive patients.

### Methods

The literature search across PubMed, ScienceDirect, BMJ Journals, ProQuest, and Springer databases yielded 419 articles. A total of 395 articles were filtered, leading to 24 articles assessed for eligibility and overall analysis, which resulted in eight included studies for quantitative synthesis. Quality appraisal used RoB 2.0, while meta-analysis used RevMan 5.4.

### Results

Our analysis results indicated significant improvements in 24-hour, daytime, and home ambulatory blood pressure measurements, favoring the uRDN over control. The mean difference (MD) for 24-hour measurements was -0.84 mmHg [95% CI -1.14, -0.55; p < 0.00001], for daytime measurements -1.27 mmHg [95% CI -1.59, -0.95; p < 0.00001], and for home measurements -1.98 mmHg [95% CI -2.32, -1.64; p < 0.00001], with moderate heterogeneity observed. Office ambulatory measurements also favored the uRDN with a

**Funding:** Airlangga University Research Funding with Grant Number: 166/UN3.LPPM/PT.01.09/2024.

**Competing interests:** The authors have declared that no competing interests exist

significant MD of -1.51 mmHg [95% CI -1.91; -1.10; p < 0.00001]. Funnel plots revealed some outliers, indicating true heterogeneity among the studies.

## Conclusion

uRDN was associated with a significant reduction (-2.32 to -0.10 mmHg) in blood pressure of hypertensive patients.

## Introduction

Chronic hypertension is a major global health challenge due to its high mortality and morbidity. Hypertension affects roughly 1.28 billion individuals, accounted for 77% of non-communicable diseases, with most cases reported in low- and middle-income countries [1]. Hypertension is responsible for slightly more than 20% of population with cardiovascular diseases, contributing to over 10 million deaths and 218 million disability-adjusted life-years globally [2, 3].

Despite various approaches for managing mild to moderate hypertension, effective management of severe or resistant hypertension remains a challenge [4]. Various pharmacological strategies have been utilized for treating hypertension, including thiazide-type diuretics, calcium channel blockers, ACE inhibitors, and ARBs [5]. However, determining the optimal antihypertensive drug dose remains a challenge [6]. While lifestyle modifications and pharmacological interventions serve as the primary pillars of hypertension therapy [7, 8], a significant number of hypertensive patients remain inadequately treated [9, 10]. Despite the standard approach of treatment for hypertension, the incidence of cardiovascular complications such as stroke, myocardial infarction, and chronic kidney disease are still rising globally. Considering the current treatment limitations, there is an urgent need to explore the interventional approach that can improve the outcome of the current standard treatment. Non-pharmacological alternatives, such as endovascular renal denervation, have emerged as promising interventions for managing hypertension [11].

The sympathetic innervation of the kidney plays a pivotal role in blood pressure regulation, which makes it an interesting therapeutic target for novel intervention [12]. Among the novel interventions, renal denervation (RDN), particularly ultrasound-based renal denervation (uRDN), emerges as a ground-breaking approach [13]. This technique, evolving from its radiofrequency-based predecessor (rRDN), harnesses acoustic energy for precise, circumferential ablation, enhanced by endovascular cooling for improved depth and completeness. This advancement symbolizes a significant leap in treating resistant hypertension and the relentless pursuit of superior therapeutic strategies to manage sympathetic nerve activity more effectively [14–16].

While earlier systematic reviews have focused on catheter-based renal denervation, the safety and efficacy of uRDN technology have not been fully explored [13]. This systematic review and meta-analysis aim to bridge this knowledge gap, offering a comprehensive assessment of uRDN's performance against traditional treatments. More than just a comparison, this study reviews whether uRDN can effectively reduce ambulatory blood pressure in hypertensive patients without the aid of antihypertensive drugs. By integrating and synthesizing existing evidence, this research seeks to provide critical insights that could revolutionize clinical practices, potentially transforming the landscape of strategy in hypertension management.

## Methods

This meta-analysis followed the Preferred Reporting Items for Systematic Reviews and Meta-Analysis protocol (PRISMA) [15]. This study was registered in **PROSPERO** with the registration number **CRD42023493100** on December 23rd, 2023.

### Study eligibility criteria

The Inclusion criteria of the studies were: 1) Randomized controlled trials (RCTs) that reported on the implementation of renal denervation with the application of ultrasound settings with sham/placebo as the comparator to highlight the efficacy of uRDN, 2) Studies whose population of resistant hypertensive patients, and 3) Ultrasound renal denervation as the intervention. After assessing the eligibility of each study, we excluded some of the studies due to; 1) Renal denervation that employed other methods, such as radiofrequency implementation for renal denervation and injection of neurolytic agents into tissue, 2) non-retrievable/incomplete studies, 3) different outcomes other than the employed parameters as it may cause bias within the data analysis.

### Search strategy

Two authors (RNR, FAG) conducted literature searches through seven databases including PubMed, Cochrane, ScienceDirect, BMJ Journals, Google Scholar, ProQuest, and Springer from December 2023 until January 2024. A randomized controlled trial filter was applied if it was available on the database search. The literature search was carried out with keywords using Boolean operators as detailed in **S1 Data in S1 File.** The inclusion criteria of this meta-analysis refer to the patient, intervention, control, outcome, time, and settings (PICOTS) framework in S2 **Data in S1 File**.

### Data extraction

Eligible studies were screened based on the inclusion criteria and eliminated based on the exclusion criteria. After evaluating the studies for eligibility, two authors (FAG and DDCHR) extracted data from the included studies. This data encompassed baseline information such as the country of the study, the age of the sample, the total number of patients, the specific number of patients in the intervention and sham-control groups, the gender distribution, and the duration of follow-up. Additionally, outcomes were extracted for each included study, focusing on the mean change in blood pressure measured through 24-hour ambulatory monitoring, as well as daytime, night time, home, and office measurements. Any discrepancies in data extraction, including variations in study parameters and denominations, were discussed and resolved with the input of the other four authors (RNR, N, SEW, and GNPJ). Included studies were assessed further with quantitative and qualitative synthesis.

### Qualitative synthesis

The risk of bias in included studies was assessed using The Revised Tool for Risk of Bias in Randomized Trials (RoB 2.0). Afterward, the results were inputted into the "bias" section of the spreadsheet. The spreadsheet was then uploaded to the ROBVIS website to display the assessment result using the traffic light system effectively.

### Quantitative synthesis

Review Manager 5.4 software (Cochrane Collaboration, Oxford, UK) was used for the meta-analysis. Clinical outcomes from continuous data were reported as mean difference (MD) and

95% confidence interval (CI) and presented using a forest plot. The $I^2$ method was used to calculate statistical heterogeneity (25% was considered low heterogeneity, 25–50% moderate heterogeneity, and >50% high heterogeneity). A random effect model was used to conduct additional analysis when the meta-analysis revealed significant heterogeneity. $I^2 > 50\%$ was considered significantly heterogeneous while $P < 0.05$ was considered statistically significant.

## Results

### Study selection and identification

Following the elimination of duplicate studies and abstract screening, a total of 24 full-text randomized controlled trials (RCTs) underwent a comprehensive evaluation. Ultimately, eight clinical trials were chosen for inclusion in the meta-analysis, as depicted in **Fig 1**. Six articles were excluded due to inaccessibility, four studies were excluded due to incomplete information, and another six were excluded because the data was not presented as full-text articles. The studies were assessed, and different outcomes were revealed to determine the efficacy of ultrasound renal denervation to several hypertension parameters, such as night-time ambulatory blood pressure, daytime ambulatory blood pressure, home ambulatory blood pressure, and office ambulatory blood pressure.

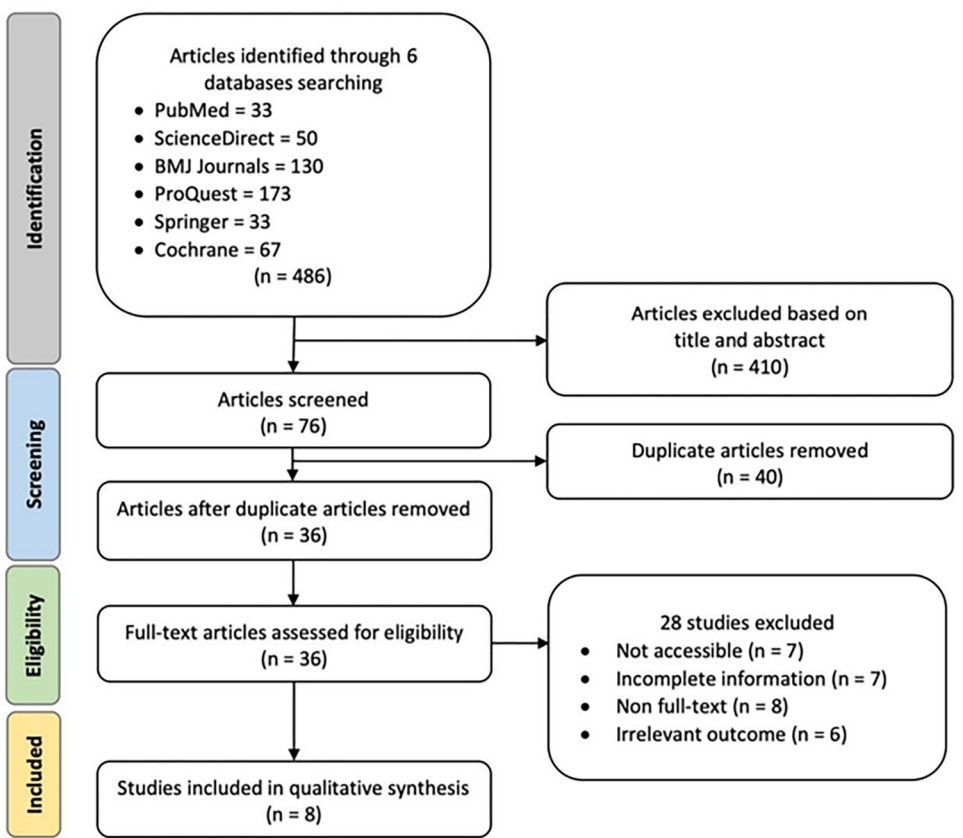

**Fig 1. Preferred reporting items for systematic reviews and meta-analyses (PRISMA) flowchart for study identification and selection.** The original database search resulted in a total of 486 studies from six databases searched, including PubMed, ScienceDirect, BMJ journals, ProQuest, Springer, and Cochrane. 410 articles were removed to prevent duplication. A total of 76 articles were filtered, resulting in 36 articles assessed for eligibility and overall analysis. 28 articles were removed due to inaccessibility, resulting in eight clinical trials in the qualitative synthesis.

## Demography and clinical characteristics of the included studies

The demography and clinical characteristics of each study were examined and listed in **S3A and S3B Data in S1 File.** The studies included were conducted in countries such as the USA, Europe, and Asian countries. Eight of the studies included in our meta-analysis utilized RCTs designs. Sample sizes ranged from 50 to 214 participants per study, with a total of 989 participants. Follow-up duration for each study ranging from two months to three years, with a higher percentage of male participants, totaling 409 males and 194 females, while the sex of the remaining participants was not stated. Eight of these studies investigate five outcomes to determine the efficacy of uRDN to several hypertension parameters, including 24-h ambulatory, night-time ambulatory blood pressure, daytime ambulatory blood pressure, home ambulatory blood pressure, and office ambulatory blood pressure. Each outcome was described in the forest plot analysis as follows.

The study conducted by Fengler et al. [17] in Germany included 50 patients with therapy-resistant hypertension, defined as SBP over 135 mmHg or DBP over 90 mmHg, as measured by ABPM, despite being on at least three antihypertensive medications, including one diuretic. The patients were evenly divided into two groups: an intervention group and a sham-controlled group, with 25 patients in each group.

The studies by Azizi et al. (2018) [18] under the RADIANCE-HTN SOLO trial were conducted across 21 centers in the USA and 18 in Europe. A total of 146 patients were randomly assigned to either the renal denervation group (n = 74) or the sham procedure group (n = 72). To evaluate the effect of uRDN, all antihypertensive medications were discontinued four weeks prior to the post-intervention ambulatory blood pressure assessment.

The trial conducted by Mahfoud et al. [16] recruited participants from multiple centers across the USA and Europe, including France, Germany, the Netherlands, Belgium, and the United Kingdom. The study included patients who met the inclusion criteria of uncontrolled hypertension, defined as office blood pressure (BP) ≥140/90 mmHg and <180/110 mmHg on 0–2 antihypertensive medications, or controlled hypertension with office BP <140/90 mmHg on 1 or 2 antihypertensive medications. In total, 72 patients were included in the study, with 33 patients crossing over to receive uRDN as the intervention.

Azizi et al.'s [19] trial was a multicenter, randomized controlled trial involving patients from 28 tertiary centers in the USA and Europe. A total of 136 patients were randomly assigned to either the renal denervation group (n = 69) or the sham procedure group (n = 67). Resistant hypertension was defined as a seated office blood pressure of at least 140/90 mmHg despite the use of three or more antihypertensive medications, including a diuretic. To evaluate the efficacy of uRDN, all eligible patients were switched to a once-daily, fixed-dose, single-pill combination of a calcium channel blocker, an angiotensin receptor blocker, and a thiazide diuretic until the specified blood pressure criteria were exceeded (180/110 mmHg for office blood pressure or 170/105 mmHg for home blood pressure).

The study by Kario et al. in 2022 [20] recruited patients from Japan and South Korea. The key inclusion criteria included adults with resistant hypertension, defined as a seated office blood pressure of ≥150/90 mmHg and a 24-hour ambulatory systolic blood pressure of ≥140 mmHg, despite being on a stable regimen that included the maximum tolerated dosages of at least three antihypertensive medications from different classes. Participants were randomized to either the uRDN group (n = 72) or the sham procedure group (n = 71).

The study conducted by Saxena et al. [21] using the Paradise catheter system (ReCor Medical, Palo Alto, CA, USA) in patients with combined systolic-diastolic hypertension. The study included male and female patients aged 18 to 75 years with office blood pressure readings of

≥140/≥90 mmHg who were on 0–2 antihypertensive medications. After a 4-week washout period without antihypertensive treatment, participants were required to have daytime ambulatory blood pressure readings between ≥135/≥85 mmHg and <170/<105 mmHg to qualify for the study. The primary outcome measure was the change in office blood pressure from baseline to 12 months, with secondary endpoints including assessments of renal function and any adverse effects.

The study by Azizi et al., 2023 [22] was a multi-center, double-blind, randomized clinical trial conducted from across 37 centers in the United States and 24 centers in Europe, with randomization stratified by center. The study involved 150 patients aged 18 to 75 years with hypertension, defined as a seated office SBP of ≥140 mm Hg and DBP of ≥90 mm Hg, despite the use of up to two antihypertensive medications. To qualify, participants were required to have an ambulatory SBP/DBP of at least 135/85 mm Hg and less than 170/105 mm Hg after a 4-week medication washout period. Additionally, eligible patients needed an eGFR of 40 mL/min/1.73 m$^2$ or greater and suitable renal artery anatomy. Participants were randomly assigned in a 2:1 ratio to either undergo ultrasound renal denervation or receive a sham procedure. During the 2-month follow-up, patients were instructed to avoid antihypertensive medications unless their blood pressure exceeded predefined criteria accompanied by clinical symptoms.

## Quality appraisal

The final clinical trial studies that were included in the analysis underwent a thorough quality assessment using the RoB 2.0. The evaluation revealed that all six included studies exhibited a consistently low risk of bias (Fig 2).

## The impact of ultrasound renal denervation on 24-hour ambulatory blood pressure monitoring

A total of eight studies were included in this meta-analysis, the safety and efficacy of URD intervention were compared to control/placebo in hypertensive patients. Eight studies reported the 24-h ambulatory with a total sample size of 526 for intervention, 400 in the control group for both SBP and DBP. As seen in **Fig 3,** the results showed a significant overall result favoring the URD with MD of -0.80 [95% CI -1.10; -0.51]. Heterogeneity was found to be moderately high and significant. The funnel plot showed a few outliners as seen in **Fig 8A** revealing true heterogeneity among the studies.

## The impact of ultrasound renal denervation on daytime ambulatory blood pressure monitoring

Eight studies reported the daytime ambulatory with a total sample size of 519 for intervention, 399 in control group for both SBP and DBP. The results as seen in **Fig 4** showed a significant overall result favoring the URD with MD of -1.19 [95% CI -1.15; -0.87]. Heterogeneity was found to be moderately high and significant. The funnel plot showed a few outliners as seen in **Fig 8B** revealing true heterogeneity among the studies.

## The impact of ultrasound renal denervation on night time ambulatory blood pressure monitoring

Eight studies reported the night-time ambulatory with a total sample size of 525 for intervention, 398 in the control group in the DBP and SBP. The results as seen in **Fig 5** showed a non-significant overall result favoring the URD groups with MD of -0.03 [95% CI -0.43;

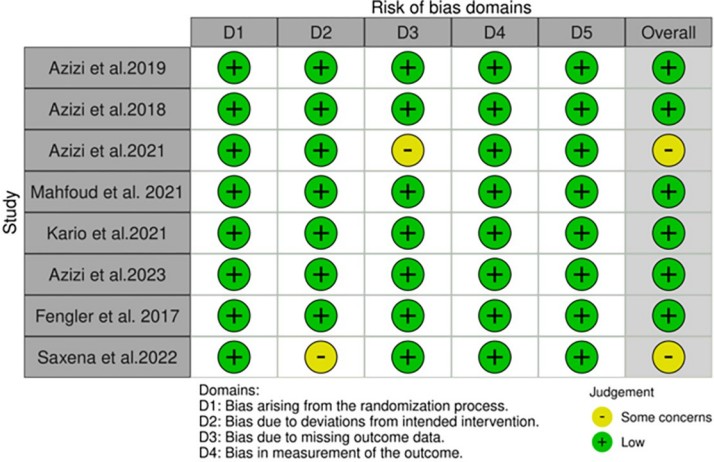

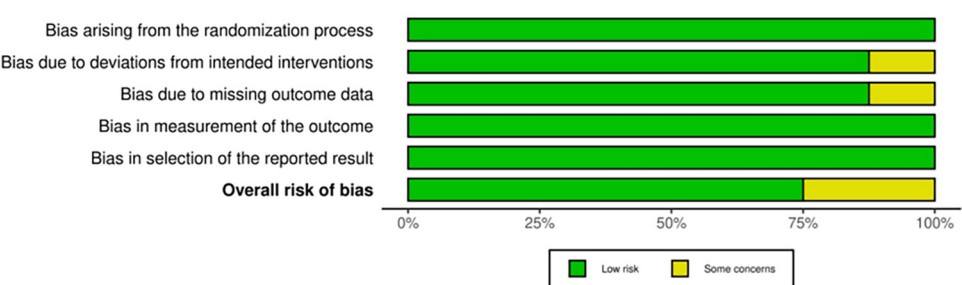

**Fig 2. Risk of bias summary using the Cochrane risk of bias 2.0 tool for randomized-controlled trial studies.** The green region represents studies with a low risk of bias, the yellow region shows studies with unclear risk of bias, and the red region shows studies with a high risk of bias.

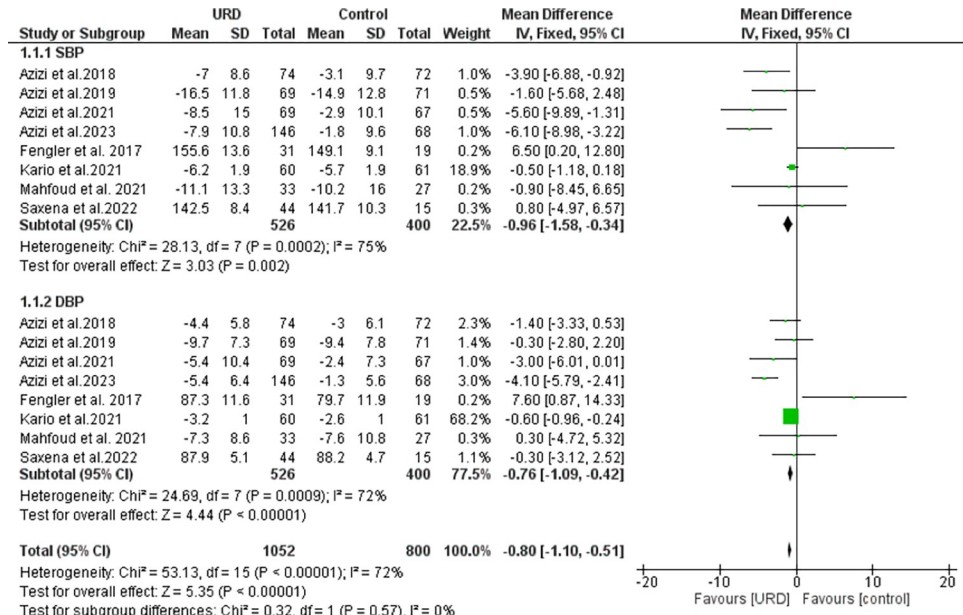

**Fig 3. Forest plot of the mean difference of 24-hour ambulatory blood pressure monitoring in uRDN compared to control.** The blue square and solid lines represent the odds ratio with 95% confidence intervals. The size of the squares indicates the weight of each study. The black rhombus indicates the pooled estimate with 95% confidence intervals.

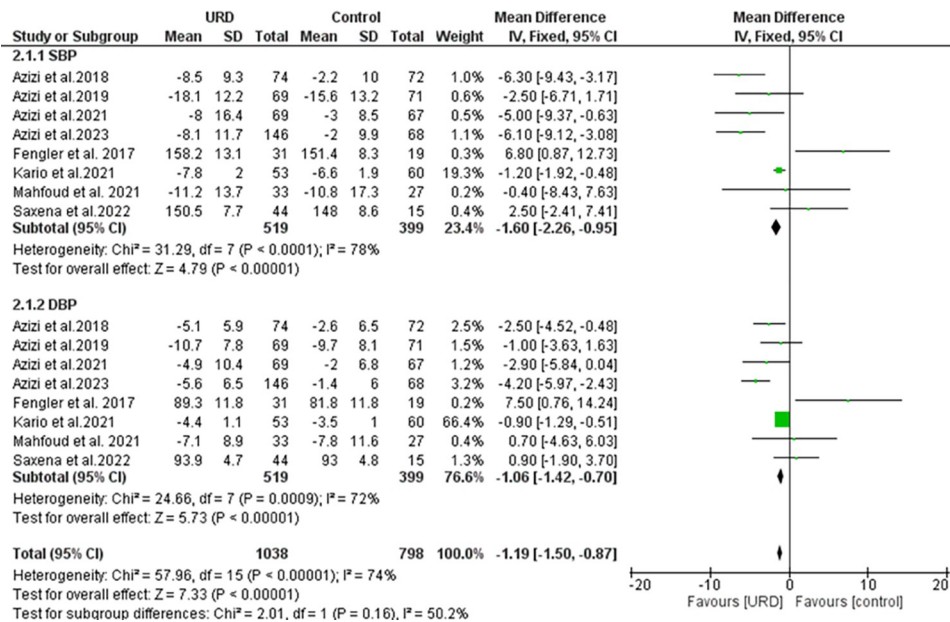

**Fig 4. Forest plot of the mean difference of daytime ambulatory blood pressure monitoring in uRDN compared to control.** The blue square and solid lines represent the odds ratio with 95% confidence intervals. The size of the squares indicates the weight of each study. The black rhombus indicates the pooled estimate with 95% confidence intervals.

0.37]. Heterogeneity was found to be moderately high and significant. The funnel plot showed just three outliners as seen in **Fig 8C** revealing true heterogeneity among the studies.

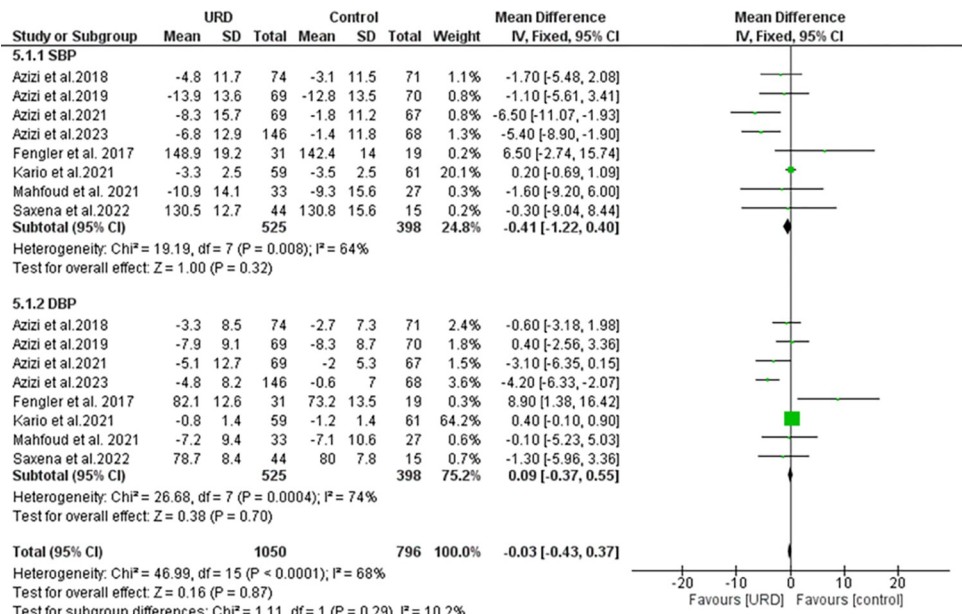

**Fig 5. Forest plot of the mean difference of night-time ambulatory blood pressure monitoring in uRDN compared to control.** The blue square and solid lines represent the odds ratio with 95% confidence intervals. The size of the squares indicates the weight of each study. The black rhombus indicates the pooled estimate with 95% confidence intervals.

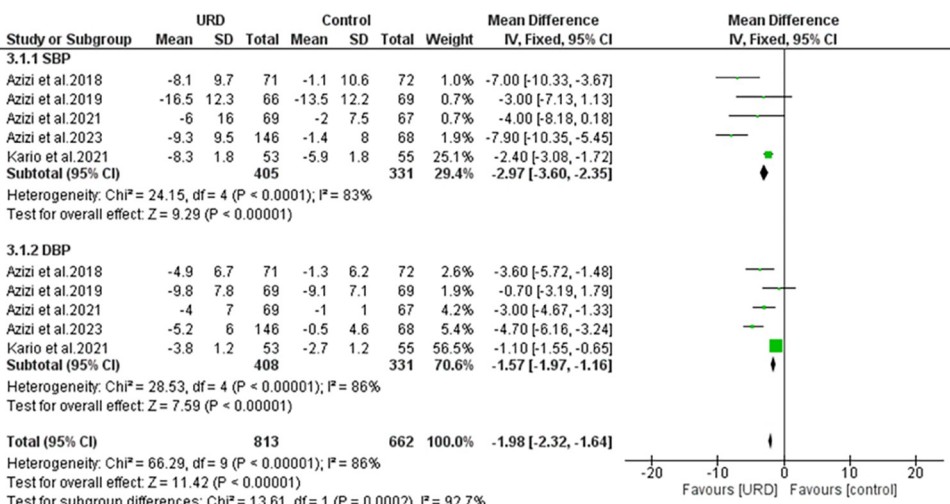

**Fig 6. Forest plot of the mean difference of home ambulatory blood pressure monitoring in uRDN compared to control.** The blue square and solid lines represent the odds ratio with 95% confidence intervals. The size of the squares indicates the weight of each study. The black rhombus indicates the pooled estimate with 95% confidence intervals.

## The impact of ultrasound renal denervation on home ambulatory blood pressure monitoring

Five studies reported the home ambulatory with a total sample size of 405 for intervention, 331 in the control group for SBP and 408 for intervention, 331 in the control group for DBP. As seen in **Fig 6**, the results showed a significant overall result the URD favouring with MD of -1.98 [95% CI -2.32; -1.64]. Heterogeneity was found to be high and significant. The funnel plot showed three outliners as seen in **Fig 8D** revealing true heterogeneity among the studies.

## The impact of ultrasound renal denervation on office ambulatory blood pressure monitoring

Eight studies reported the Office ambulatory with a total sample size of 524 for intervention, 398 in the control group for both SBP and DBP. The results as seen in **Fig 7** showed a significant overall result favoring the URD with MD of -1.40 [95% CI -1.81; -1.00]. Heterogeneity was found to be moderately high and significant. The funnel plot showed few outliners as seen in **Fig 8E** revealing true heterogeneity among the studies. The publication of bias analysis for each parameter can be seen in **Fig 8**.

## Meta-regression analysis on each outcome measurement

Noting the high heterogeneity seen throughout assessed outcomes, we perform a meta-regression analysis with the mean difference of each ambulatory blood pressure measurement as the dependent variable and the participants' gender and sample sizes as covariates (**S5 and S6 Data in S1 File**). The meta-regression analysis reveals that both gender and sample size are significant moderators of the effect sizes in ambulatory blood pressure measurements. Gender differences lead to increased blood pressure readings in 24-hour, nighttime, and home settings, while larger sample sizes correlate with increased blood pressure readings across all evaluated settings (24-hour, daytime, home, and office).

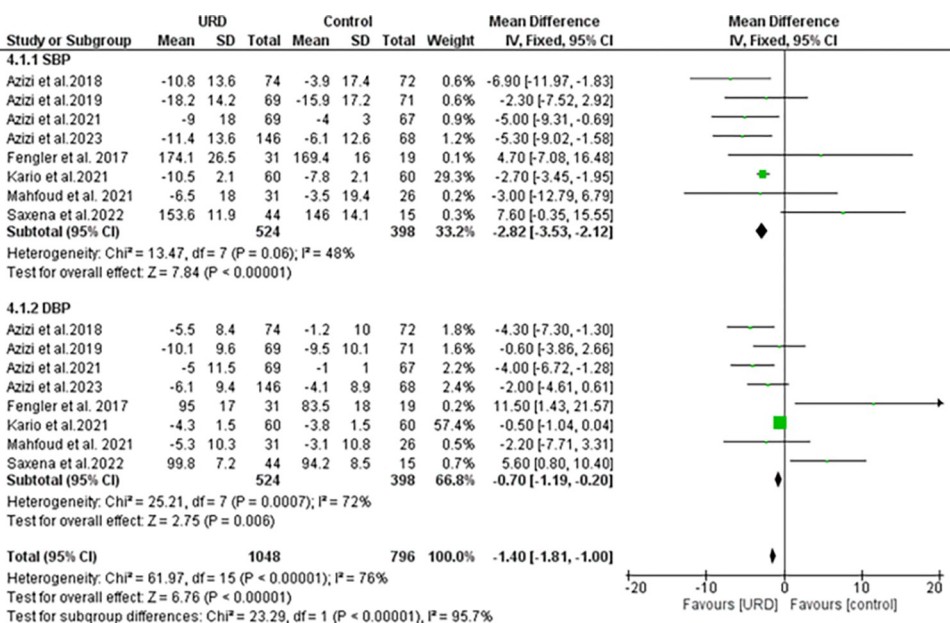

**Fig 7. Forest plot of the mean difference of office ambulatory blood pressure monitoring in uRDN compared to control.** The blue square and solid lines represent the odds ratio with 95% confidence intervals. The size of the squares indicates the weight of each study. The black rhombus indicates the pooled estimate with 95% confidence intervals.

## Discussion

The renal denervation procedure for treating hypertension incorporates three methods including ultrasound application, radiofrequency energy utilization, and the injection of neurolytic agents into the vascular tissues surrounding the kidneys [23]. Three significant multicenter, international, blinded, randomized, sham-controlled trials have evaluated ultrasound-based renal denervation. The RADIANCE-HTN SOLO trial investigated endovascular ultrasound renal denervation in patients who had stopped using anti-hypertensive drugs [14]. The RADIANCE-HTN TRIO trial focused on patients resistant to a combination pill of three anti-hypertensive drugs [19]. The RADIANCE II trial assessed the technique in patients with high blood pressure despite taking up to two anti-hypertensive drugs [22]. Notably, the RADIANCE-HTN TRIO trial demonstrated blood pressure reduction in patients unresponsive to various medications, highlighting the procedure's efficacy in resistant hypertension. Across these studies, 506 individuals aged 18 to 75 were involved, emphasizing the potential of uRDN in hypertension management.

The endovascular ultrasound catheter, designed for deployment in the main renal arteries, offers a distinct advantage over multi-electrode radiofrequency catheters [17]. While positioned before the arterial bifurcation, it directs ablative energy uniformly around the artery's circumference, up to 1 mm below the luminal surface. This method, despite utilizing fewer ablations compared to radiofrequency techniques, has achieved similar blood pressure reductions [24]. The uRDN is typically performed under conscious sedation and involve multiple standard practice for analgesia and anticoagulation, including the use of unfractionated heparin. A unique aspect of the Paradise system is the balloon-tipped catheter, delivering high-frequency ultrasound energy for precise, ring-shaped ablation, while sparing the proximal tissue. This procedure, involving 2–3 ultrasound emissions per artery, has been executed with meticulous care regarding patient safety, including post-procedure management [25].

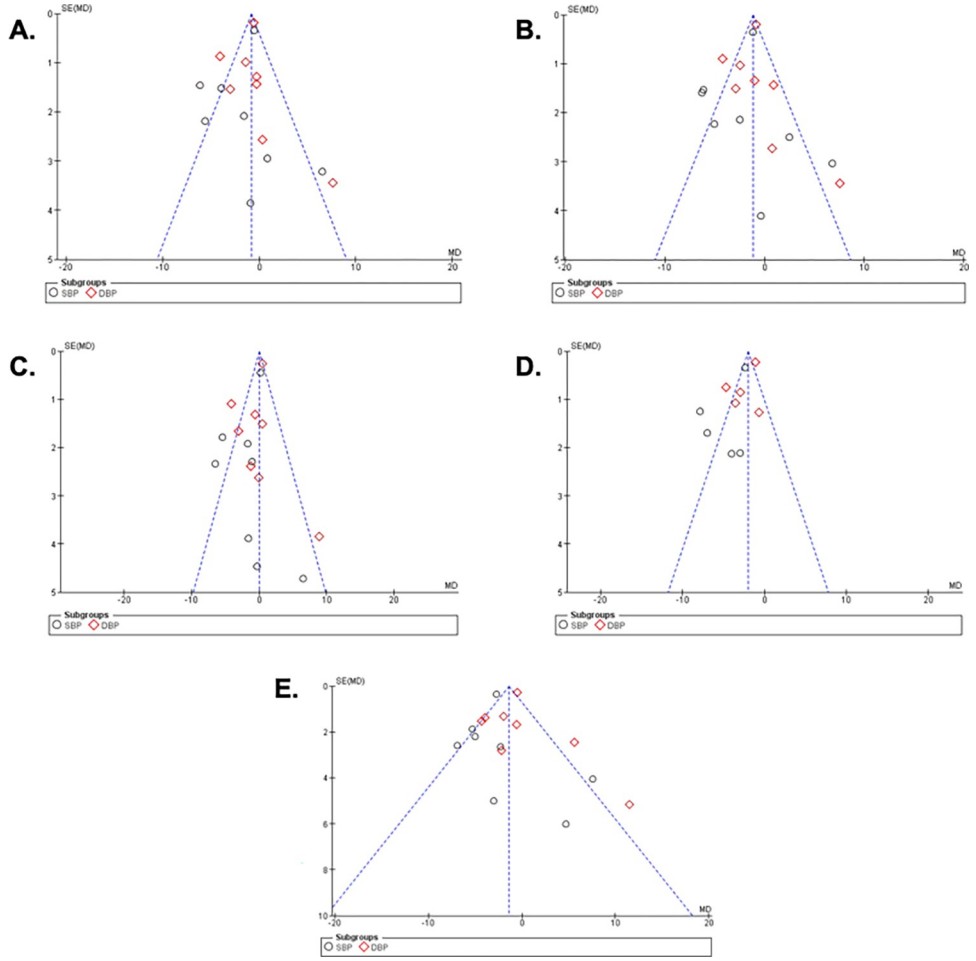

**Fig 8. Funnel plots of included studies.** Funnel plots of: A.24-hour Ambulatory Blood Pressure Monitoring; B. Daytime Ambulatory Blood Pressure Monitoring; C. Night-time Ambulatory Blood Pressure Monitoring; D. Home Ambulatory Blood Pressure Monitoring; and E. Office Ambulatory Blood Pressure Monitoring.

The efficacy of uRDN is further evidenced by its consistent impact on blood pressure control across various monitoring settings, such as 24-hour, daytime, home, and office ambulatory BP monitoring (ABPM). Notably, uRDN maintains safety, with no significant adverse events or major complications, including renal artery stenosis, reported in most patients [22, 25]. These findings corroborate previous studies confirming the safety of uRDN, aligning with the general safety characteristics observed in both ultrasound and radiofrequency-based renal denervation techniques [14, 19, 26, 27]. This compelling evidence highlights uRDN as an effective and safe intervention for hypertension management.

Recent findings consistently demonstrate the effectiveness of uRDN in BP management. A meta-analysis of three randomized clinical trials showed a significant reduction in daytime ambulatory systolic BP with uRDN compared to controls at a 2-month follow-up [28]. This analysis revealed a more pronounced decrease in systolic BP among uRDN-treated patients (mean difference [MD] = -8.5 mmHg) than those undergoing sham procedures (MD = -2.2 mmHg). Notably, uRDN led to reductions in various BP measurements, including daytime and 24-hour ambulatory systolic and diastolic pressures and office and home readings [14]. Further supporting this, a recent randomized controlled trial (RCT) found the uRDN group

experienced a notably greater reduction in daytime ambulatory systolic BP (MD = -7.9 mmHg) compared to the sham group (MD = -1.8 mmHg), amounting to a 6.3 mmHg greater decrease [22]. These consistent and substantial reductions across different BP measurements underscore uRDN's potential as an effective treatment for hypertension.

Another meta-analysis reported marginal decreases in blood pressure (BP), but these were not statistically significant for 24-hour ambulatory systolic BP (mean difference [MD] = -2.15 mmHg; p = 0.15), diastolic BP (MD = -1.19 mmHg; p = 0.18), or office systolic BP (MD = -1.71 mmHg; p = 0.67) [29]. The lack of significant results might stem from the small number of studies in this analysis. Our study expanded the pool to eight studies to address this, offering a more robust assessment. Additionally, our study observed inconsistent effects on night-time ABPM, potentially due to physiological night-time dipping in BP influenced by circadian rhythms, which can obscure differences in night-time BP measurements [30]. However, the minimal change in mean difference in decreasing BP across other BP measurements under-scores the general trend of uRDN's effectiveness in lowering BP in patients with resistant hypertension.

Hypertension, often labelled as the 'silent killer' for its asymptomatic character, continues to pose a significant challenge in global health, with its prevalence alarmingly rising each year [31]. In this context, the introduction of the Paradise Renal Denervation System by ReCor Medical in Palo Alto, CA, marks a significant innovation. Diverging from previous radiofre-quency renal denervation devices, this system offers enhanced nerve coverage and reduces operator dependency, potentially improving outcomes for hypertensive patients. The system's ultrasound-based technology, notable for its efficacy and safety, allows for procedures averag-ing just 20 minutes. Its ability to achieve ablation depths of 1 to 6 mm increases the precision of nerve targeting, thereby decreasing the likelihood of adverse events and complications [25].

The implementation of uRDN has shown promising results in reducing and stabilizing blood pressure in hypertensive patients. This technique presents a valuable alternative for patients who are noncompliant with medication, ensuring that treatment efficacy is not hin-dered. When combined with lifestyle modifications, such as regular exercise and dietary adjustments, uRDN could significantly alter the landscape of hypertension management [19]. Previous study indicate that adding uRDN to standard of care is a cost-effective treatment strategy for patients with resistant hypertension, with an incremental cost-effectiveness ratio (ICER) of £5600 (6500), assuming the long-term durability and safety of uRDN effects observed in the RADIANCE-HTN TRIO trial [19]. Modeling showed a greater than 99% prob-ability that this approach is cost-effective in the UK, based on a willingness-to-pay (WTP) threshold of £20,000 (23,214) [32]. This conclusion was consistent across various sensitivity and scenario analyses, all yielding ICERs below this threshold. The validity of the model was confirmed against previous economic models [33]. Earlier cost-effectiveness analyses have demonstrated that the Symplicity radiofrequency renal denervation system is a cost-effective resource allocation [33–35]. uRDN could be feasible for implementation in clinical settings as it offers a non-invasive or minimally invasive alternative for patients with resistant hyperten-sion who have not responded well to traditional medications. This can be particularly benefi-cial for patients who may be reluctant to undergo more invasive procedures. Additionally, clinical studies have shown promising results in terms of efficacy, safety, and cost-effectiveness, further supporting its feasibility as a treatment option [19, 35, 36].

However, it is important to recognize the limitations of current research. The lack of exten-sive RCTs in this area is a notable drawback, highlighting the need for more in-depth research to strengthen the findings and enhance the reliability and depth of conclusions. Another limi-tation of this study is the exclusion of non-English language studies, which could introduce biases and limitations. This exclusion may lead to language bias, potentially overlooking

valuable insights and findings from studies conducted in languages other than English and limiting the diversity and inclusivity of the data pool. It also restricts the generalizability of the findings and may hinder a comprehensive understanding of the research topic, as healthcare practices and perspectives vary globally. Additionally, the impact of publication bias, as indicated by funnel plot outliers, suggests a need for deeper analysis and potential adjustments in meta-analysis results to address such biases and ensure the study's robustness and reliability.

In summary, the uRDN system represents a major advancement in the treatment of hypertension. Its demonstrated efficacy and safety profile positions it as a potentially transformative tool in addressing this widespread health issue. However, the need for further research, particularly through rigorous RCTs, remains critical. Such studies are essential to vialidate this innovative approach's benefits and solidify its role in the broader context of hypertension management. As the medical community continues to seek effective solutions for this pervasive condition, the potential of technologies like uRDN offers a beacon of hope for patients worldwide.

## Conclusion

In conclusion, utilizing uRDN demonstrated a notable association with a substantial reduction in blood pressure among hypertensive patients across various settings, albeit with discernible variability observed between individual studies.

## Supporting information

**S1 File. Supplementary data 1 to 6.** (1) Literature search terms, (2) PICOTS framework, (3A) list of examined studies, (3B) Demography and Clinical Characteristics of The Included Studies, (4) PRISMA Checklist, (5) Meta-regression Analysis of Gender on All Outcomes, and (6) Meta-regression Analysis of Sample Size on All Outcomes, and (7) Protocol for handling missing data.
(DOCX)

**S2 File. The efficacy and safety of ultrasound renal denervation on hypertensive patients: A meta-analysis of clinical cases.**
(DOCX)

## Acknowledgments

We thank Dr. Ika Kadariswantiningsih (Airlangga University) for the technical advice during this study.

## Author Contributions

**Conceptualization:** Roy Novri Ramadhan, Derren David Christian Homenta Rampengan, Maulana Antiyan Empitu.

**Formal analysis:** Roy Novri Ramadhan, Derren David Christian Homenta Rampengan.

**Funding acquisition:** Maulana Antiyan Empitu.

**Investigation:** Hiroyuki Yamada, Satriyo Dwi Suryantoro, Maulana Antiyan Empitu.

**Methodology:** Roy Novri Ramadhan, Derren David Christian Homenta Rampengan, Felicia Angelica Gunawan, Satriyo Dwi Suryantoro, Maulana Antiyan Empitu.

**Project administration:** Felicia Angelica Gunawan.

**Resources:** Satriyo Dwi Suryantoro.

**Supervision:** Roy Novri Ramadhan, Hiroyuki Yamada, Mochammad Thaha, Satriyo Dwi Suryantoro, Maulana Antiyan Empitu.

**Validation:** Felicia Angelica Gunawan, Satriyo Dwi Suryantoro, Maulana Antiyan Empitu.

**Visualization:** Maulana Antiyan Empitu.

**Writing – original draft:** Roy Novri Ramadhan, Derren David Christian Homenta Rampengan, Felicia Angelica Gunawan, Nathania, Sebastian Emmanuel Willyanto.

**Writing – review & editing:** Roy Novri Ramadhan, Nathania, Sebastian Emmanuel Willyanto, Maulana Antiyan Empitu.

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
