## [Decision Letter · Decision Letter 0]

15 May 2024

PONE-D-24-12822Ultrasound Renal Denervation in Hypertensive Patients: A Systematic Review and Meta-AnalysisPLOS ONE

Dear Dr. Empitu,

Thank you for submitting your manuscript to PLOS ONE. After careful consideration, we feel that it has merit but does not fully meet PLOS ONE’s publication criteria as it currently stands. Therefore, we invite you to submit a revised version of the manuscript that addresses the points raised during the review process.

**ACADEMIC EDITOR: **All issues raised by expert reviewers are required.

We look forward to receiving your revised manuscript.

Kind regards,

Vincenzo Lionetti, M.D., PhD

Academic Editor

PLOS ONE

Journal Requirements:

"Airlangga University Research Funding with Grant Number: 166/UN3.LPPM/PT.01.09/2024"

3. Please upload a copy of Figures 9 and 10, to which you refer in your text on page 8. If the figure is no longer to be included as part of the submission please remove all reference to it within the text.

5. We notice that your supplementary [Supplementary Tables 1 and 2] are included in the manuscript file. Please remove them and upload them with the file type 'Supporting Information'. Please ensure that each Supporting Information file has a legend listed in the manuscript after the references list.

Reviewers' comments:

Reviewer's Responses to Questions

**Comments to the Author**

1. Is the manuscript technically sound, and do the data support the conclusions?

Reviewer #1: Yes

Reviewer #2: Yes

2. Has the statistical analysis been performed appropriately and rigorously? 

Reviewer #1: Yes

Reviewer #2: No

3. Have the authors made all data underlying the findings in their manuscript fully available?

Reviewer #1: Yes

Reviewer #2: Yes

4. Is the manuscript presented in an intelligible fashion and written in standard English?

Reviewer #1: Yes

Reviewer #2: Yes

5. Review Comments to the Author

Reviewer #1: The manuscript titled "Ultrasound Renal Denervation in Hypertensive Patients: A Systematic Review and Meta-Analysis" aims to evaluate the safety and efficacy of ultrasound renal denervation (uRDN) in managing hypertension. This manuscript provides a thorough analysis of the available literature on uRDN in hypertension. Please find below some specific comments and suggestions for improvement.

The authors use a comprehensive range of databases, but there's no mention of grey literature or hand-searching reference lists, which can often uncover additional relevant studies. Including these sources could potentially alter the conclusions and should be considered to ensure a comprehensive review.

The manuscript lacks detailed search strings and configurations for each database. For reproducibility and to allow peers to assess any potential biases or omissions in the search strategy, detailed search terms, date limits, and filters applied should be clearly provided.

While the authors outline their criteria, they do not sufficiently justify why certain types of studies were excluded or why non-English studies were omitted, risking cultural and publication biases. This exclusion could skew the findings towards a particular demographic or healthcare system, limiting the generalizability of the results.

The manuscript could improve transparency by detailing the data extraction process more explicitly. For instance, describing how discrepancies between reviewers were resolved and whether any form of calibration was done before starting the full review would lend more credibility to the reliability of the extracted data.

The paper acknowledges moderate to high heterogeneity but lacks a rigorous exploration or explanation of its potential sources. The authors should conduct subgroup analyses or meta-regressions to explore how different patient populations, uRDN techniques, or study designs might influence outcomes.

The manuscript reports statistically significant findings with minimal mean differences in blood pressure that may not translate into significant clinical benefits. It is crucial to discuss the clinical relevance of these findings in the context of existing treatments, considering both the magnitude of effect and the risks or costs associated with uRDN.

The discussion should more critically address the potential biases introduced by the exclusion of non-English language studies and the inclusion criteria. Moreover, the impact of publication bias, as suggested by the funnel plot outliers, should be analyzed more deeply, potentially adjusting the meta-analysis results for such bias.

The manuscript should discuss the characteristics of the populations studied in the included trials in more depth, such as age ranges, severity of hypertension, and previous treatments. This would help readers understand who the findings apply to and under what circumstances uRDN might be most effective.

While the manuscript adds to the literature by consolidating data on uRDN, the practical implications of implementing these findings in clinical settings are not thoroughly explored. Discussion around the feasibility, cost, patient acceptance, and comparison with existing hypertension treatments would be valuable.

The paper could propose more specific future research needs, such as the need for longer-term outcome studies, comparative effectiveness research against other hypertension interventions, and trials focused on patient-reported outcomes to better understand the impact of uRDN on quality of life.

Reviewer #2: This paper presents a contemporary systematic review and meta-analysis of RCTs of RDN.

Although an interesting and generally well reported and conducted study there are number of important issues that require attention

1. review strategy - why were web of science and Embase not searched?

2. results - the interpretation of the findings of these trials is very dependent on the precise nature of the recruited populations. The results need a much more detailed description of the population of each study. Suggest that population, intervention, comparator and outcomes for included RCTs is tabulated. The study populations need to consider in the interpretation of the results of the review and the conclusions

3. data analysis - suggest as well present Funnel plots, also perform Egger testing for asymmetry. The methods describe the authors will use random effects model in the presence of stats significant heterogeneity. However, all Forest plots use fixed effect meta-analysis

6. PLOS authors have the option to publish the peer review history of their article (what does this mean?). If published, this will include your full peer review and any attached files.

Reviewer #1: No

Reviewer #2: **Yes: **Prof Rod Taylor

---

## [Author Response · Author response to Decision Letter 0]

21 Jul 2024

RESPONSE TO REVIEWER QUESTION

EDITOR COMMENTS:

Response: Thank you for the suggestion. We have carefully ensured that our manuscript complies with PLOS ONE's style requirements and followed the provided templates for formatting. All aspects have been formatted according to PLOS ONE guidelines. We have also adhered to the specified file naming conventions to ensure consistency and clarity in our submission.

2. Thank you for stating the following financial disclosure: "Airlangga University Research Funding with Grant Number: 166/UN3.LPPM/PT.01.09/2024" Please state what role the funders took in the study. If the funders had no role, please state: "The funders had no role in study design, data collection and analysis, decision to publish, or preparation of the manuscript." If this statement is not correct you must amend it as needed. Please include this amended Role of Funder statement in your cover letter; we will change the online submission form on your behalf.

Response: We confirm that this research did not receive any specific grant from funding agencies in the public, commercial, or not-for-profit sectors.

3. Please upload a copy of Figures 9 and 10, to which you refer in your text on page 8. If the figure is no longer to be included as part of the submission please remove all reference to it within the text.

Response: Our apologies for the mistake in the caption. The text has been revised and we thank you for your concern.

Response: We have included captions for all Supporting Information files at the end of the manuscript and updated any in-text citations accordingly. We have also removed Supplementary Tables 1 and 2 from the manuscript file and uploaded them as 'Supporting Information' with legends listed in the manuscript after the references list.

5. We notice that your supplementary [Supplementary Tables 1 and 2] are included in the manuscript file. Please remove them and upload them with the file type 'Supporting Information'. Please ensure that each Supporting Information file has a legend listed in the manuscript after the references list.

Response: We have removed Supplementary Tables 1 and 2 from the main manuscript file and uploaded them separately as 'Supporting Information'. Each Supporting Information file now has a legend listed in the manuscript after the references list, as requested.

REVIEWERS COMMENTS:

Reviewer #1: 

The manuscript titled "Ultrasound Renal Denervation in Hypertensive Patients: A Systematic Review and Meta-Analysis" aims to evaluate the safety and efficacy of ultrasound renal denervation (uRDN) in managing hypertension. This manuscript provides a thorough analysis of the available literature on uRDN in hypertension. Please find below some specific comments and suggestions for improvement.

1. The authors use a comprehensive range of databases, but there's no mention of grey literature or hand-searching reference lists, which can often uncover additional relevant studies. Including these sources could potentially alter the conclusions and should be considered to ensure a comprehensive review.

Response: Thank you for the encouraging comments and constructive suggestions. We acknowledge the importance of including grey literature and hand-searching reference lists. We have updated our methods section to include a search of grey literature, including the Cochrane database (Page 5). This additional search has been detailed in the revised manuscript.

2. The manuscript lacks detailed search strings and configurations for each database. For reproducibility and to allow peers to assess any potential biases or omissions in the search strategy, detailed search terms, date limits, and filters applied should be clearly provided.

Response: Thank you for the suggestion. Detailed search terms, date limits, and filters applied for each database have been added to the Supporting Information files (Page 1). This information will enhance the reproducibility of our study and allow peers to assess any potential biases or omissions in our search strategy.

3. While the authors outline their criteria, they do not sufficiently justify why certain types of studies were excluded or why non-English studies were omitted, risking cultural and publication biases. This exclusion could skew the findings towards a particular demographic or healthcare system, limiting the generalizability of the results.

Response: We have added a detailed justification for excluding certain types of studies. After assessing the eligibility of each study, we excluded some of the studies due to; 1) Renal denervation that employed other methods, such as radiofrequency implementation for renal denervation and injection of neurolytic agents into tissue, 2) non-retrievable/incomplete studies, 3) different outcomes other than the employed parameters as it may cause bias within the data analysis. This justification addresses potential cultural and publication biases and discusses how these exclusions might affect the generalizability of our results.

4. The manuscript could improve transparency by detailing the data extraction process more explicitly. For instance, describing how discrepancies between reviewers were resolved and whether any form of calibration was done before starting the full review would lend more credibility to the reliability of the extracted data.

Response: The data extraction process has been elaborated upon in the methods section (Page 5). We describe how discrepancies between reviewers were resolved and the calibration process conducted before starting the full review to ensure the reliability of the extracted data. Any discrepancies in data extraction, including variations in study parameters and denominations, were discussed and resolved with the input of the other four authors (RNR, N, SEW, and GNPJ) by detailing the data extraction process more explicitly. This data encompassed baseline information such as the country of the study, the age of the sample, the total number of patients, the specific number of patients in the intervention and sham-control groups, the gender distribution, and the duration of follow-up. Additionally, outcomes were extracted for each included study, focusing on the mean change in blood pressure measured through 24-hour ambulatory monitoring, as well as daytime, nighttime, home, and office measurements. 

5. The paper acknowledges moderate to high heterogeneity but lacks a rigorous exploration or explanation of its potential sources. The authors should conduct subgroup analyses or meta-regressions to explore how different patient populations, uRDN techniques, or study designs might influence outcomes

Response: Thank you for pointing out this issue. We have conducted meta-regressions to explore potential sources of heterogeneity. These analyses are included in the revised manuscript, and we discuss how the differences in patient populations, uRDN techniques, or study designs might influence outcomes (Page 9). The meta-regression analysis reveals that both gender and sample size are significant moderators of the effect sizes in ambulatory blood pressure measurements. Gender differences lead to increased blood pressure readings in 24-hour, nighttime, and home settings, while larger sample sizes correlate with increased blood pressure readings across all evaluated settings (24-hour, daytime, home, and office).

6. The manuscript reports statistically significant findings with minimal mean differences in blood pressure that may not translate into significant clinical benefits. It is crucial to discuss the clinical relevance of these findings in the context of existing treatments, considering both the magnitude of effect and the risks or costs associated with uRDN.

Response: The discussion section now includes an analysis of the clinical relevance of our findings, considering the magnitude of the effect and the risks or costs associated with uRDN compared to existing treatments (Page 12).

7. The discussion should more critically address the potential biases introduced by the exclusion of non-English language studies and the inclusion criteria. Moreover, the impact of publication bias, as suggested by the funnel plot outliers, should be analyzed more deeply, potentially adjusting the meta-analysis results for such bias.

Response: We have addressed the potential biases introduced by excluding non-English language studies and our inclusion criteria (Page 12). The impact of publication bias, as suggested by the funnel plot outliers, has been analyzed more deeply, and we have adjusted the meta-analysis results accordingly. Based on our analysis results, gender differences result in higher blood pressure measurements during 24-hour monitoring, nighttime, and at home. Additionally, larger sample sizes are associated with elevated blood pressure readings in all examined environments, including 24-hour, daytime, home, and office settings.

8. The manuscript should discuss the characteristics of the populations studied in the included trials in more depth, such as age ranges, severity of hypertension, and previous treatments. This would help readers understand who the findings apply to and under what circumstances uRDN might be most effective.

Response: This information has been disclosed in the table of characteristics in Supporting Information file (Page 3). The characteristics of the populations studied in the included trials, such as age ranges, severity of hypertension, and previous treatments, are now discussed in greater depth (Page 3). This information will help readers understand the applicability of our findings.

9. While the manuscript adds to the literature by consolidating data on uRDN, the practical implications of implementing these findings in clinical settings are not thoroughly explored. Discussion around the feasibility, cost, patient acceptance, and comparison with existing hypertension treatments would be valuable.

Response: We have expanded the discussion to explore the practical implications of implementing our findings in clinical settings, including feasibility, cost, patient acceptance, and comparison with existing hypertension treatments (Page 12). uRDN could be feasible for implementation in clinical settings as it offers a non-invasive or minimally invasive alternative for patients with resistant hypertension who have not responded well to traditional medications. Additionally, clinical studies have shown promising results in terms of efficacy, safety, and cost-effectiveness, further supporting its feasibility as a treatment option (Azizi et al., 2015; Azizi et al., 2021; Taylor et al., 2024)

10. The paper could propose more specific future research needs, such as the need for longer-term outcome studies, comparative effectiveness research against other hypertension interventions, and trials focused on patient-reported outcomes to better understand the impact of uRDN on quality of life.

Response: Future research needs have been proposed (Page 13), including the need for longer-term outcome studies, comparative effectiveness research against other hypertension interventions, and trials focused on patient-reported outcomes to better understand the impact of uRDN on quality of life.

Reviewer #2: 

This paper presents a contemporary systematic review and meta-analysis of RCTs of RDN. Although an interesting and generally well reported and conducted study there are number of important issues that require attention

1. review strategy - why were web of science and Embase not searched?

Response: Thank you for the constructive suggestion. We agree that supplementing PubMed search with Embase can increase the study coverage by 5-6%. We did not use Embase search to accompany the PubMed search due to the access limitation to the database, which requires a subscription not covered by our institution and grants. To increase our search coverage, we supplemented the PubMed search with other general databases such as Google Scholar, ScienceDirect, and ProQuest.

2. results - the interpretation of the findings of these trials is very dependent on the precise nature of the recruited populations. The results need a much more detailed description of the population of each study. Suggest that population, intervention, comparator and outcomes for included RCTs is tabulated. The study populations need to consider in the interpretation of the results of the review and the conclusions

Response: Thank you for the suggestion. A detailed description of the population of each study, including population, intervention, comparator, and outcomes (PICO), has been added in “Supporting Information” files (Page 2). This information is tabulated in the table of characteristics and PICOs table (Page 1 and 2). 

3. data analysis - suggest as well present Funnel plots, also perform Egger testing for asymmetry. The methods describe the authors will use random effects model in the presence of stats significant heterogeneity. However, all Forest plots use fixed effect meta-analysis

Response: Thank you for your suggestion. In our analysis, we used a fixed effects model despite the presence of statistically significant heterogeneity. We observed that the weights assigned to each study were relatively equal. To address the heterogeneity, we also tried to apply the random effects model, but significant heterogeneity persisted. To highlight the high heterogeneity in our outcomes, we have included funnel plots for all outcomes in our presentation. Additionally, we conducted meta-regression to further support our findings.

---

## [Decision Letter · Decision Letter 1]

6 Aug 2024

PONE-D-24-12822R1Ultrasound Renal Denervation in Hypertensive Patients: A Systematic Review and Meta-AnalysisPLOS ONE

Dear Dr. Empitu,

Thank you for submitting your manuscript to PLOS ONE. After careful consideration, we feel that it has merit but does not fully meet PLOS ONE’s publication criteria as it currently stands. Therefore, we invite you to submit a revised version of the manuscript that addresses the points raised during the review process.

**ACADEMIC EDITOR: **Some relevant issues remain to be addressed. Point-by-point revision is required.

We look forward to receiving your revised manuscript.

Kind regards,

Vincenzo Lionetti, M.D., PhD

Academic Editor

PLOS ONE

Journal Requirements:

Reviewers' comments:

Reviewer's Responses to Questions

**Comments to the Author**

1. If the authors have adequately addressed your comments raised in a previous round of review and you feel that this manuscript is now acceptable for publication, you may indicate that here to bypass the “Comments to the Author” section, enter your conflict of interest statement in the “Confidential to Editor” section, and submit your "Accept" recommendation.

Reviewer #1: All comments have been addressed

Reviewer #2: (No Response)

2. Is the manuscript technically sound, and do the data support the conclusions?

Reviewer #1: Yes

Reviewer #2: No

3. Has the statistical analysis been performed appropriately and rigorously? 

Reviewer #1: Yes

Reviewer #2: Yes

4. Have the authors made all data underlying the findings in their manuscript fully available?

Reviewer #1: Yes

Reviewer #2: Yes

5. Is the manuscript presented in an intelligible fashion and written in standard English?

Reviewer #1: Yes

Reviewer #2: (No Response)

6. Review Comments to the Author

Reviewer #1: The Authors have revised their manuscript according to my comments. I think that the manuscript has improved and I have no more comments.

Reviewer #2: whilst the authors have responded to most comment appropriately, I am not persuaded that they addressed the issue about adequately summarising the nature of the different populations. There is still no narrative text in the results section and there should be. I can see the supplemental table that reviewers say includes the description of the population description to judge if adequate. Detailed population description is a key aspect of the interpretation of this review so I would wish to see these details before I can accept,

7. PLOS authors have the option to publish the peer review history of their article (what does this mean?). If published, this will include your full peer review and any attached files.

Reviewer #1: No

Reviewer #2: No

---

## [Author Response · Author response to Decision Letter 1]

28 Aug 2024

Dr. Emily Chenette

Editor-in-Chief

Journal of PLOS One

Dear Dr. Emily Chenette,

I am writing to submit a revised version of our manuscript entitled “Ultrasound Renal Denervation in Hypertensive Patients: A Systematic Review and Meta-Analysis” to PLOS One. We greatly appreciate the constructive feedback provided by the reviewers and have made significant revisions to the manuscript to address their concerns.

We have carefully considered all the reviewers' comments and made several changes accordingly. We have elaborated in narration text regarding the characteristics of each study in more detail in our main manuscript. The changes and additional data have been outlined in the attached response to reviewers. Additionally, we have included a highlighted version of the manuscript to indicate the revisions made.

These changes have significantly improved the manuscript and addressed the reviewers' concerns. We hope you will find this revised version suitable for publication in PLOS One.

Sincerely,

Maulana A. Empitu, M.D., Ph.D.

Faculty of Medicine, Airlangga University, Surabaya, Indonesia

Email: maulana.antiyan@fk.unair.ac.id

RESPONSE TO REVIEWER QUESTION

EDITOR COMMENTS:

1. Whilst the authors have responded to most comments appropriately, I am not persuaded that they addressed the issue about adequately summarizing the nature of the different populations. There is still no narrative text in the results section and there should be. I can see the supplemental table that reviewers say includes the description of the population description to judge if adequate. 

Thank you for your thorough review and for pointing out the need for a more detailed narrative regarding the characteristics of the different populations in our study. We have included a more comprehensive narrative in the results section of our main manuscript (page 6-9). This narrative text elaborates on the characteristics of each study population in greater detail, addressing the diversity and specific attributes of the populations involved. Additionally, we have ensured that this information is now directly accessible in the main text and not only the supplemental tables (page 6-9). We appreciate your advice in improving the clarity and comprehensiveness of our manuscript.

---

## [Decision Letter · Decision Letter 2]

16 Sep 2024

Ultrasound Renal Denervation in Hypertensive Patients: A Systematic Review and Meta-Analysis

PONE-D-24-12822R2

Dear Dr. Empitu,

We’re pleased to inform you that your manuscript has been judged scientifically suitable for publication and will be formally accepted for publication once it meets all outstanding technical requirements.

Kind regards,

Vincenzo Lionetti, M.D., PhD

Academic Editor

PLOS ONE

Additional Editor Comments (optional):

Reviewers' comments:

Reviewer's Responses to Questions

**Comments to the Author**

1. If the authors have adequately addressed your comments raised in a previous round of review and you feel that this manuscript is now acceptable for publication, you may indicate that here to bypass the “Comments to the Author” section, enter your conflict of interest statement in the “Confidential to Editor” section, and submit your "Accept" recommendation.

Reviewer #2: All comments have been addressed

2. Is the manuscript technically sound, and do the data support the conclusions?

Reviewer #2: Yes

3. Has the statistical analysis been performed appropriately and rigorously? 

Reviewer #2: Yes

4. Have the authors made all data underlying the findings in their manuscript fully available?

Reviewer #2: Yes

5. Is the manuscript presented in an intelligible fashion and written in standard English?

Reviewer #2: Yes

6. Review Comments to the Author

Reviewer #2: authors in my opinion have responded to other reviewer's concern. Happy to approve this paper for acceptance

7. PLOS authors have the option to publish the peer review history of their article (what does this mean?). If published, this will include your full peer review and any attached files.

Reviewer #2: No

---

## [Editor Report · Acceptance letter]

28 Nov 2024

PONE-D-24-12822R2 

PLOS ONE

Dear Dr. Empitu, 

I'm pleased to inform you that your manuscript has been deemed suitable for publication in PLOS ONE. Congratulations! Your manuscript is now being handed over to our production team.

Kind regards, 

on behalf of

Prof. Vincenzo Lionetti 

Academic Editor

PLOS ONE